# Distributional Depth-Based Estimation
# of Object Articulation Models

**Ajinkya Jain**[*]
UT Austin

**Stephen Giguere**[†]
UT Austin

**Rudolf Lioutikov**[†]
Karlsruhe Institute of Technology

**Scott Niekum**
UT Austin

**Abstract:** We propose a method that efficiently learns distributions over articulation model parameters directly from depth images without the need to know articulation model categories a priori. By contrast, existing methods that learn articulation models from raw observations typically only predict point estimates of the model parameters, which are insufficient to guarantee the safe manipulation of articulated objects. Our core contributions include a novel representation for distributions over rigid body transformations and articulation model parameters based on screw theory, von Mises-Fisher distributions, and Stiefel manifolds. Combining these concepts allows for an efficient, mathematically sound representation that implicitly satisfies the constraints that rigid body transformations and articulations must adhere to. Leveraging this representation, we introduce a novel deep learning based approach, DUST-net, that performs category-independent articulation model estimation while also providing model uncertainties. We evaluate our approach on several benchmarking datasets and real-world objects and compare its performance with two current state-of-the-art methods. Our results demonstrate that DUST-net can successfully learn distributions over articulation models for novel objects across articulation model categories, which generate point estimates with better accuracy than state-of-the-art methods and effectively capture the uncertainty over predicted model parameters due to noisy inputs. [webpage]

**Keywords:** Articulated Objects, Model Learning, Uncertainty Estimation

## 1   Introduction

Articulated objects, such as drawers, staplers, refrigerators, and dishwashers, are ubiquitous in human environments. These objects consist of multiple rigid bodies connected via mechanical joints such as hinge joints or slider joints. Robots in human environments will need to interact with these objects often while assisting humans in performing day-to-day tasks. To interact safely with such objects, a robot must reason about their articulation properties while manipulating them. An ideal method for learning such properties might estimate these parameters directly from raw observations, such as RGB-D images while requiring limited or no a priori information about the task. The ability to additionally provide a confidence over the estimated properties, would allow such a method to be leveraged in the development of safe motion policies for articulated objects [1].

The majority of existing methods to learn articulation models for objects from visual data either need fiducial markers to track motion between object parts [2–5] or require textured objects [6–10]. Recent deep-learning based methods address this by predicting articulation properties for objects from raw observations, such as depth images [11–14] or PointCloud data [15, 16]. However, the majority of these methods [11, 12, 15, 16] require knowledge of the articulation model category for the object (e.g., whether it has a revolute or prismatic joint) which may not be available in many realistic settings. Alleviating this requirement, Jain et al. [14] introduced ScrewNet, which uses a unified representation based on screw transformations to represent different articulation types and performs category-independent articulation model estimation directly from raw depth images. However, ScrewNet [14] and related methods [11–13, 15, 16] only predict point estimates for an object's

---

[*]Corresponding author: `ajinkya@utexas.edu`
[†]Equal contribution, presented alphabetically

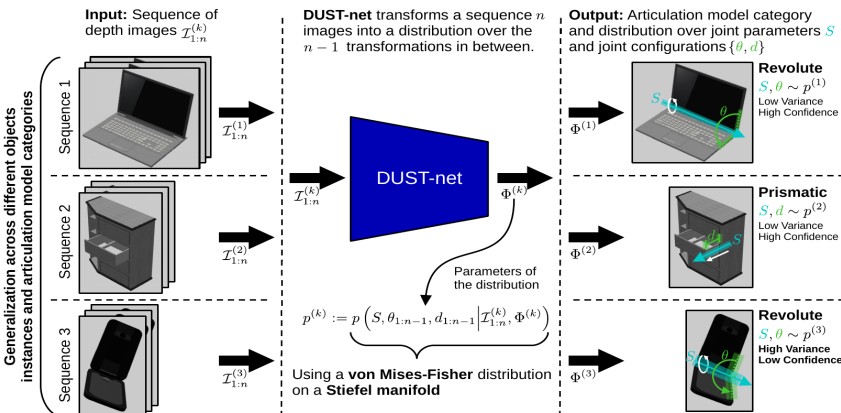

Figure 1: DUST-net uses a sequence of images $\mathcal{I}_{1:n}$ to compute the parameters, $\Phi$, of the conditional distribution over the joint parameters $S$ and configurations $\{\theta, d\}_{1:n-1}$. This distribution allows for inference and reasoning, such as uncertainty and confidence, over both the parameters and the configurations. Using a von Mises-Fisher distribution on a Stiefel manifold allows for an efficient reparameterization that inherently obeys multiple constraints that define rigid body transformations.

articulation model parameters. Nonetheless, reasoning about the uncertainty in the estimated parameters can provide significant advantages for ensuring success in robot manipulation tasks, and allows for further advancements such as robust planning [1], active learning using human queries [17], and the learning of behavior policies that provide safety assurances [18]. Motivated by these advantages, we propose a method for learning articulation models, which estimates the uncertainty over model parameters using a novel distribution over the set of screw transformations based on the matrix von Mises-Fisher distribution over Stiefel manifolds [19]. We introduce DUST-net, **D**eep **U**ncertainty estimation on **S**crew **T**ransforms-**net**work, a novel deep learning-based method that, in addition to providing point estimates of the object's articulation model parameters, leverages raw depth images to provide uncertainty estimates that can be used to guide the robot's behavior without requiring to knowledge of the object's articulation model category a priori.

DUST-net garners numerous benefits over existing methods. First, DUST-net estimates articulation properties for objects with uncertainty estimates, unlike most current methods [11–16]. These uncertainty estimates, apart from helping robots to manipulate objects safely [1], could allow a robot to take information-gathering actions when it is not confident and enhance its chances of success in completing the task. Second, similar to ScrewNet [14], DUST-net can estimate model parameters without the need to to know the articulation model category a priori, by leveraging the unified representation for different articulation model types. Third, this unified representation helps DUST-net to be more computationally and data-efficient than other state-of-the-art methods [11, 12], as it uses a single network to estimate model parameters for all common articulation models, unlike other methods that require a separate network for each articulation model category [11, 12, 15, 16]. Empirically, DUST-net outperforms other methods even when trained using only half the training data in comparison. Fourth, the distributional learning setting yields more robustness to outliers and noise. Fifth, DUST-net is able to reliably estimate distributions over articulation model parameters for objects in the robot's camera frame. By contrast, ScrewNet [14], the most closely related approach to ours, can only predict point estimates for articulation model parameters in the object's local frame.

We evaluate DUST-net through experiments on two benchmarking datasets: a simulated articulated objects dataset [11] and the PartNet-Mobility dataset [20–22], as well as three real-world objects: a microwave, a drawer, and a toaster oven. We compare DUST-net with two state-of-the-art methods, namely ScrewNet [14] and an MDN-based method proposed by Abbatematteo et al. [11], as well as two baseline methods. The experiments demonstrate that the samples drawn from the distributions learned by DUST-net result in significantly better estimates for articulation model parameters in comparison to the point estimates predicted by other methods. Additionally, the experiments show that DUST-net can successfully and accurately capture the uncertainty over articulation model parameters resulting from noisy inputs.

## 2 Related Work

**Articulation model estimation from visual observations:** A widely used approach for estimating articulation models is based on the probabilistic framework proposed by Sturm et al. [2]. It uses the time-series observations of 6D poses of different parts of an articulated object to learn the relationship between them [2, 5, 6, 10]. More recently, Abbatematteo et al. [11] and Li et al. [12] proposed methods to learn articulation properties for objects from raw depth images given articulation model category. In a related body of work on object parts mobility estimation, Wang et al. [15] and Yan et al. [16] proposed approaches to segment different parts of the object in an input point cloud and estimate their mobility relationships, given a known articulation model category. Alleviating the requirement of having a known articulation model category, Jain et al. [14] recently proposed ScrewNet that performs category-independent articulation model estimation from depth images. However, these methods only predict point estimates for the articulation model parameters, while DUST-net predicts a distribution over their values.

**Rigid Body Pose Estimation**: Our contributions are related to existing work on estimating distributions describing the orientation of rigid bodies. Gilitschenski et al. [23], Arun Srivatsan et al. [24], Srivatsan et al. [25] and Rosen et al. [26] propose strategies that can be used to estimate the rigid body transformation of an object using a combination of Bingham and Gaussian distributions, and the von Mises-Fisher distribution, respectively. The mathematical model used by our approach is inspired by these works, but 1) extends them to also represent uncertainty over the configuration of articulated object components about screw axes, and 2) integrates them into a deep learning model that is capable of learning these configurations from raw depth images. In addition, while these approaches use distributions over orientations and rigid body transformations to produce estimates, DUST-net directly outputs a distribution that can be used to facilitate further applications such as uncertainty-aware behavior planning.

**Interactive perception (IP)**: Katz and Brock [3] introduced IP as a method to leverage a robot's interaction with objects to generate a rich perceptual signal for articulation model estimation for planar objects, and extended it to learn 3D articulation models for objects [4]. Martín-Martín et al. [8] used hierarchical recursive Bayesian filters to make estimation more robust and developed online methods for articulation model estimation from RGB images [7–9]. A comprehensive survey on IP methods in robotics was presented by Bohg et al. [27]. While IP presents a powerful tool for estimating articulation properties for objects, a wide majority of existing IP methods require textured objects, unlike DUST-net, which learns these properties using depth images.

**Further approaches**: Articulation motion models can be viewed as geometric constraints imposed on multiple rigid bodies. Such constraints can be learned from human demonstrations by leveraging different sensing modalities [13, 28–31]. Recently, Daniele et al. [30] proposed a multimodal learning framework that incorporates both vision and natural language information for articulation model estimation. However, these approaches predict point estimates for the articulation model parameters, unlike DUST-net, which predicts a distribution over the articulation model parameters.

## 3 Problem Formulation:

Given a sequence of $n$ depth images $\mathcal{I}_{1:n}$ of motion between two parts of an articulated object, we estimate the parameters of a probability distribution $p(\phi|\mathcal{I}_{1:n})$ representing uncertainty over the parameters $\phi$ of the articulation model $\mathcal{M}$ governing the motion between the two parts. Following Jain et al. [14], we define the model parameters $\phi$ as the parameters of the screw axis of motion, $\mathsf{S} = (\mathbf{l}, \mathbf{m})$, where both $\mathbf{l}$ and $\mathbf{m}$ are elements of $\mathbb{R}^3$. This unified parameterization can be used in articulation models with at most one degree-of-freedom (DoF), namely rigid, revolute, prismatic, and helical [14]. Additionally, we estimate the parameters of a distribution $p(q_{1:n-1}|\mathcal{I}_{1:n})$ representing uncertainty over the configurations $q_{1:n-1}$ identifying the rigid body transformations between the two parts in the given sequence of images $\mathcal{I}_{1:n}$ under model $\mathcal{M}$ with parameters $\phi$. Configurations $q_i, i \in \{1...n-1\}$ correspond to a set of tuples, $q_i = (\theta_i, d_i)$, defining a rotation around and a displacement along the screw axis $\mathsf{S}^3$. We assume that the relative motion between the two object parts is determined by a single articulation model.

---

[3]Please refer to the supplementary material for further details

## 4 Approach

Given a sequence of depth images $\mathcal{I}_{1:n}$ of motion between two parts of an articulated object, DUST-net estimates parameters of the joint probability distribution $p(\phi, q_{1:n-1}|\mathcal{I}_{1:n})$ representing uncertainty over the articulation model parameters $\phi$ governing the motion between the two parts and the observed configurations $q_{1:n-1}$. When deciding how to learn this distribution, two goals arise. While some parameters, such as the translation of an object part along a screw axis, are defined on Euclidean space, the set of valid screw axes exhibits constraints that prevent standard distributions defined on $\mathbb{R}^6$ from being applied without complicating the learning process. For example, a standard representation for distributions over screw axes can be the product of a Bingham distribution over the line's orientation and a multivariate normal distribution over its position in space [32]. However, this representation produces non-unique estimation targets. A rotation of $\theta$ about the screw axis with orientation $l$ results in the same transformation as a rotation of $-\theta$ about the screw axis with orientation $-l$. Similarly, a displacement $d$ along $l$ results in the same transformation as a displacement $-d$ along $-l$. This leads to ambiguities in the targets in the estimation problem and can hinder the performance of the trained estimator. By selecting a representation that accounts for these symmetries, these non-unique estimation targets are removed. Second, once a suitable parameterization is chosen, we seek a parametric form for the joint distribution which can be learned by a deep network.

First, we consider the problem of parameterizing the set of screw axes. As noted earlier, we define the model parameter $\phi$ as the parameters of the screw axis of motion $S = (l, m)$. However, this parameterization requires that $l$ has unit norm, and that $l$ and $m$ are orthogonal. To eliminate these constraints, we rewrite the moment vector of a screw axis as $m = \|m\|\,\hat{m}$, where $\|m\|$ and $\hat{m}$ represent its magnitude and a unit vector along it respectively, and the Plücker coordinates for the screw axis as $S = (l, \hat{m}, \|m\|)$. The Plücker coordinates can then be seen as an unconstrained point in the space $\mathbb{S} := V_{2,3} \times \mathbb{R}^+$, where $(l, \hat{m}) \in V_{2,3}$ with $V_{2,3}$ denoting the *Stiefel manifold* of 2-frames in $\mathbb{R}^3$ and $\|m\| \in \mathbb{R}^+$ with $\mathbb{R}^+$ denoting the set of positive real numbers. The Stiefel manifold $V_{k,m}$ is the space whose points are sets of $k$ orthonormal vectors in $\mathbb{R}^m$, called $k$-frames in $\mathbb{R}^m$ $(k \leq m)$[1] [19]. Consequently, because of the one-to-one mapping from elements of $V_{2,3} \times \mathbb{R}^+$ to screw axes, the non-unique estimation targets described above are eliminated. Based on this parametrization of screw axes, we define the set of valid configuration parameters as follows. We restrict the range of values for the rotation about the screw axis to be $\theta \in [0, 2\pi)$ and restrict the displacement along the axis to be $d \in \mathbb{R}^+$. Note that these constraints do not reduce the representational power of the screw transform $(l, m, \theta, d)$ to denote a general rigid body transform, but merely ensure a unique representation.

Having described the parameterization of the set of screw axes and configurations, we now consider the task of defining a joint probability distribution over their values. We propose to represent the distribution over predicted screw axis parameters, $p(S \mid \mathcal{I}_{1:n})$ with $S \in \mathbb{S}$, as a product of a matrix von Mises-Fisher distribution $\mathcal{F}(\cdot|3, F)$ defined on the Stiefel manifold $V_{2,3}$[1] and a truncated normal distribution $\mathcal{N}^+(\cdot|\mu, \sigma)$ with truncation interval $[0, +\infty)$ over $\mathbb{R}^+$. Formally,

$$p(S \mid \mathcal{I}_{1:n}) = p\left(l, \hat{m}, \|m\| \mid \mathcal{I}_{1:n}, F, \mu_m, \sigma_m^2\right) = \mathcal{F}\left(l, \hat{m} \mid 3, F\right) \mathcal{N}^+\left(\|m\| \mid \mu_m, \sigma_m^2\right), \quad (1)$$

where $F$ is a $3 \times 2$ matrix representing the parameters of the matrix von Mises-Fisher distribution over $V_{2,3}$, and $\mu_m$ and $\sigma_m$ denote the mean and standard deviation of the truncated normal distribution.

Given the sequence of $n$ images, we also wish to estimate the posterior over configurations $q_{1:n-1} = \{\theta_{1:n-1}, d_{1:n-1}\}$ corresponding to the rotations about and displacements along the screw axis $S$. We define the joint posterior representing the uncertainty over the screw axis $S$ and the configurations $\{\theta_{1:n-1}, d_{1:n-1}\}$ about it as a product of the aforementioned distribution and a set of distributions defined over the configuration parameters,

$$p(S, \theta_{1:n-1}, d_{1:n-1} \mid \mathcal{I}_{1:n}, \Phi) = p(S; F, \mu_m, \sigma_m^2)\, \Psi(\theta_{1:n-1}; \psi)\, \Upsilon(d_{1:n-1}; \upsilon) \quad (2)$$

where $\Phi = \{F, \mu_m, \sigma_m^2, \psi, \upsilon\}$ is the set of parameters for the distribution and $\Psi$ and $\Upsilon$ represent the set of distributions having parameters $\psi$ and $\upsilon$ over the configurations $\theta_{1:n-1}$ and $d_{1:n-1}$, respectively. For the sake of brevity, we present further details on modeling assumptions in the supplementary material (see Appendix B). In this work, we consider $\Psi$ and $\Upsilon$ to be products of truncated normal distributions such that $\Psi = \prod_{i=1}^{n-1} \mathcal{N}^+(\theta_i | M_\theta^i, \sigma_\theta^2)$ and $\Upsilon = \prod_{i=1}^{n-1} \mathcal{N}^+(d_i | M_d^i, \sigma_d^2)$ with

---

[1]Please refer to the supplementary material for further details

$\mathbf{M}_\theta = \{\mu_\theta^1, ..., \mu_\theta^{n-1}\}$, $\mathbf{M}_d = \{\mu_d^1, ..., \mu_d^{n-1}\}$, $\sigma_\theta$, and $\sigma_d$ denoting the set of means and the standard deviations of the set of truncated normal distributions over the configurations $\theta_{1:n-1}$ and $d_{1:n-1}$, respectively.

**Distribution parameter matrix F:** The parameter matrix for the matrix von Mises-Fisher distribution over $V_{3,2}$ is a $3 \times 2$ matrix, $\mathbf{F}$. This presents two possible parameterizations of the matrix: first, to estimate each of the 6 elements of the $3 \times 2$ matrix $\mathbf{F}$ and second, to estimate the matrices $\Gamma, \Lambda$, and $\Omega$ defining the SVD of $\mathbf{F}$, given by $\mathbf{F} = \Gamma \Lambda \Omega^T$. The second parameterization decouples the two objectives of distribution mode alignment with the ground truth labels and uncertainty representation; the mode of the distribution is given by $M = \Gamma \Omega^T$, and the concentration matrix for the distribution is given by $K = \Omega \Lambda \Omega^T$. This decoupling allows the network to independently optimize both objectives, whereas in the first parameterization, changes in the elements of $\mathbf{F}$ causes changes in both components.

By definition, $\Lambda$ is a $2 \times 2$ diagonal matrix with two independent parameters, and $\Omega \in O(2)$ is a rotation matrix in two dimensions with one independent parameter, the rotation angle $\omega$. The matrix $\Gamma \in \tilde{V}_{3,2}$ can be constructed from a rotation matrix $R \in O(3)$ by keeping only the first two columns of R. Hence, the matrix $\Gamma$ can be defined by three independent Euler angles, $(\alpha, \beta, \gamma)$ denoting rotation according to the $ZYX$ convention in the rotating frame. Euler angles can suffer from the problem of gimble lock [32], which we resolve by restricting the Euler angles to be in the ranges $\alpha \in [0, 2\pi), \beta \in [0, \pi)$, and $\gamma \in [0, 2\pi)$.

**Normalization factor:** One of the main challenges of using the matrix von Mises-Fisher distribution is the calculation of its normalization factor $_0F_1(\frac{m}{2}, \frac{1}{4}\Lambda^2)$, which is a hypergeometric function of matrix argument [19]. In this work, we approximate this hypergeometric function using a truncated series in terms of zonal polynomials, which are multivariate symmetric homogeneous polynomials and form a basis of the space of symmetric polynomials [19]. Through our preliminary experiments, we found that this truncated series is a good approximation of $_0F_1$ as it converges to a finite value, if the singular values of the $F$, i.e. $\lambda_1$ and $\lambda_2$ are less than $\lambda_{max} = 50$.

**Architecture:** DUST-net sequentially connects a ResNet-18 CNN [33] and a 2-layer MLP. ResNet-18 extracts task-relevant features from the input images, which are used by the MLP to predict a set of parameters $\Phi$ for the distribution $p(\mathsf{S}, \theta_{1:n-1}, d_{1:n-1} \mid \mathcal{I}_{1:n}, \Phi)$. The network is trained end-to-end, with ReLU activations for the hidden fully-connected layers. The first four output (out of 40) of the last linear layer of MLP, corresponding to the parameters $(\alpha, \beta, \gamma)$ and $\omega$ representing the matrices $\Gamma$ and $\Omega$ respectively, are fed through a ReLU-6 layer to ensure that the predictions map to their respective ranges. Remaining output is fed through a Softplus layer for non-negative output. Detailed network architecture is presented in the appendix (Fig. 7).

**Training:** The training data for the model consists of sequences of depth images of objects parts moving relative to each other and the corresponding screw transforms $\mathbf{y} = (l, \hat{\mathbf{m}}, \|\mathbf{m}\|, \theta_{1:n-1}, d_{1:n-1})$. The objects and depth images are rendered in Mujoco [34]. We train DUST-net by maximizing the log-probability of the labels $\mathbf{y}$ under the distribution $p(\mathbf{y} \mid \mathcal{I}_{1:n}, \Phi)$: $\mathcal{L}(\mathbf{y}, \Phi) = -\log p(\mathbf{y} \mid \Phi)$. We assume that the observed configurations in $\mathcal{I}_{1:n}$ share the same variance. We use the precision parameters rather than the standard deviations, $\sigma_{\mathbf{m}}, \sigma_\theta$ and $\sigma_d$ to represent the distribution during training for better numerical stability. Following the discussion on training MDNs by Makansi et al. [35], we separate the training in three stages. In the first stage, we assume the dispersion of the matrix von Mises-Fisher distribution to be fixed with $\Lambda = \mathrm{diag}(\lambda_0, \lambda_0)$, $\lambda_0 = 1$ and learn parameters corresponding to $\Gamma$ and $\Omega$ matrices. In the second stage, we fix the $\Lambda$ matrix and learn the rest of the parameters in the set $\Phi$. Finally, we train to predict the complete set $\Phi$.

## 5 Experiments

In this section, we evaluate DUST-net on its ability to learn articulation model parameters and uncertainty estimates. We conducted three sets of experiments evaluating DUST-net's performance under different criteria: (1) how accurate point estimates of the articulation model parameters drawn from DUST-net's estimated distribution are in comparison to the existing methods, (2) how effectively DUST-net captures the uncertainty over parameters arising from noisy input, and (3) how effectively DUST-net transfers from simulation to a real-world setting. We evaluated DUST-net's performance on two simulated benchmarking datasets: the objects dataset provided by Abbatematteo et al. [11], and the PartNet-Mobility dataset [20–22], as well as a set of three real-world

objects. From the simulated articulated object dataset [11], we considered the cabinet, microwave, and toaster oven for revolute articulations and the drawer object class for prismatic articulations. From the PartNet-Mobility dataset [20–22], we considered five object classes: the dishwasher, oven, and microwave object classes for the revolute articulation model category, and the storage furniture object class consisting of either a single column of drawers or multiple columns of drawers, for the prismatic articulation model category. Among the three sets of experiments, we conducted the first two sets of experiments on the simulated datasets, while the last set of experiments were conducted on the real-world object dataset. In all the experiments, we assumed that the input depth images are semantically segmented and contain non-zero pixels corresponding only to the two objects between which we wish to estimate the articulation model.

We compared DUST-net's performance in estimating point estimates for articulation model parameters with two state-of-the-art methods, ScrewNet [14] and an MDN-based approach proposed by Abbatematteo et al. [11]. ScrewNet estimates the object's articulation model parameters in a local frame located at the center of the object, whereas DUST-net does so directly in the camera frame. We compare our method with ScrewNet predicting parameters both in the object local frame and the camera frame. Additionally, we propose two baseline methods that estimate distributions over articulation model parameters and compare to them DUST-net. The first baseline method (vm-SoftOrtho) can be viewed as an extension of ScrewNet to a distributional setting. It represents the uncertainty over the screw axis orientation vector $\mathbf{l}$ and the direction of moment vector $\hat{\mathbf{m}}$ using two independent von Mises-Fisher distributions and imposes a soft orthogonality constraint over the modes of the two distributions. The distributions over the moment vector magnitude $\|\mathbf{m}\|$ and configurations $q_{1:n-1}$ are considered to be normal distributions. This method suffers from the same drawback as ScrewNet, i.e., the use of a soft orthogonality constraint during training, and therefore cannot predict a valid set of screw axis parameters directly, unlike DUST-net. The second baseline method (Direct $F$) uses the same probability distribution as DUST-net to represent the uncertainty over the articulation model parameters, but estimates the individual elements of the $\mathbf{F}$ matrix directly. As a result, it fails to capture the uncertainty over model parameters accurately.

## 5.1   Accuracy of Point Estimates

The first set of experiments evaluated DUST-net's accuracy in predicting point estimates for articulation model parameters. We use the mode of the estimated distribution as the point estimate for model parameters. We used two metrics to evaluate accuracy: Mean Absolute (Angular) Deviation (MAAD) and Screw Loss (Metric proposed in ScrewNet [14]). MAAD metric indicates how close the individual screw parameters are to targets, whereas the Screw Loss indicates how close the complete predicted screw transforms is to the target transforms. The MAAD metric calculates the angular distance between the orientation of the predicted and ground-truth axis orientation vectors $\mathbf{l}$ and the orientation vectors of the screw axis moment vectors $\hat{\mathbf{m}}$. For the remaining parameters ($\|\mathbf{m}\|$, $\theta_{1:n-1}$, $d_{1:n-1}$), it calculates the mean absolute deviation between the predicted and ground-truth values. The screw loss reports the angular distance between the predicted and ground-truth screw axis orientation vectors $\mathbf{l}$ as orientation error and the length of the shortest perpendicular be-

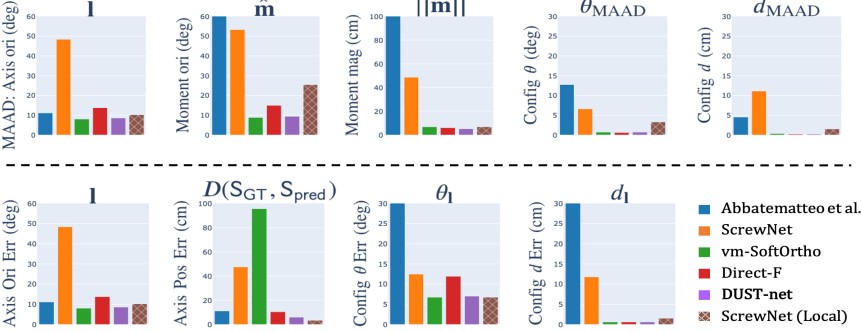

Figure 2: Mean error values on the MAAD (top) and Screw Loss (bottom) metrics for the simulated articulated objects dataset [11] (lower values are better). Point estimates for DUST-net (violet) correspond to the modes of the distributions predicted by DUST-net.

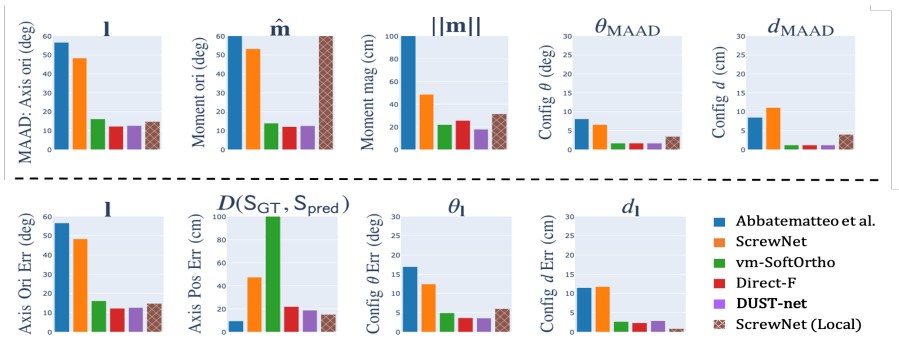

Figure 3: Mean error values on the MAAD (top) and Screw Loss (bottom) metrics for the PartNet-Mobility dataset [20–22] (lower values are better). Point estimates for DUST-net (violet) correspond to the modes of the distributions predicted by DUST-net.

tween the predicted and ground-truth screw axes as the distance between them. Configuration errors $\theta_{1:n-1}$ are reported as the difference between the predicted rotation about the predicted screw axis and the true rotation, whereas errors over $d_{1:n-1}$ are calculated as the Euclidean distance between the points displaced by the predicted and true displacements along respective axes.

Results for the synthetic articulated objects dataset and the PartNet-Mobility dataset are shown in Figures 2 and 3, respectively. Results demonstrate that under both metrics, the estimates obtained from DUST-net are typically more accurate than those obtained from the state-of-the-art methods. The first baseline, vm-SoftOrtho, performs comparably with DUST-net on both datasets when only MAAD estimates are considered. However, Figures 2 and 3 show that it produces a very high distance ($\approx$ 1m) between the predicted and ground-truth screw axes. This error arises due to the soft-orthogonality constraint used by vm-SoftOrtho, as DUST-net and the second baseline method, both of which handle the constraint implicitly, do not report high errors on that metric. Meanwhile, the second baseline, Direct $F$, performs comparably with DUST-net on both metrics for both datasets, but fails to capture the uncertainty over parameters with the required accuracy.

## 5.2 Uncertainty Estimation

The second set of experiments evaluated how effectively DUST-net's predicted distribution captures epistemic uncertainty over the predicted articulation parameters. We evaluate this by adding artificial noise to the training labels from the two simulated datasets while training DUST-net. As more noise is added, we expect the confidence estimates produces by DUST-net to decrease as well. We add noise to the labels by sampling perturbations from a matrix von Mises-Fisher distribution with varying singular values $\lambda_1$ and $\lambda_2$ of the distribution parameter matrix $\mathbf{F}$ and the truncated normal distributions with varying precision parameters $\beta_j, j \in \{\|\mathbf{m}\|, \theta, d\}$. Figure 4 show the variation of the mean of the singular values of the predicted distribution concentration matrices over screw axes by DUST-net with injected noise. In the noiseless case, the singular values of the matrix von Mises-Fisher distribution increases until they reach their maximum allowed value at $\lambda_{max} = 50$. When label noise is added,

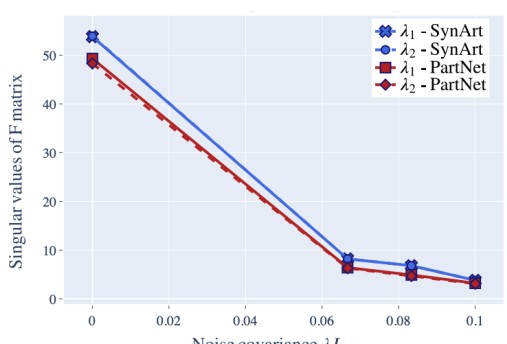

Figure 4: Variation of the mean of the singular values of predicted distribution concentration matrices over screw axes by DUST-net with artificially injected noise. Predicted singular values decrease monotonically with input noise, showing that the network's confidence over the predicted parameters decreases with input noise.

our results show that DUST-net's confidence over its predicted parameters decreases monotonically as more noise is added to the labels, supporting our hypothesis.

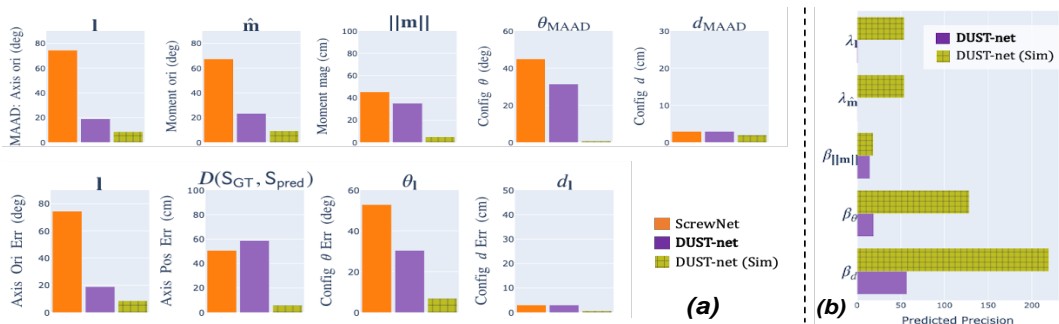

Figure 5: **(a)** Mean error values on MAAD (top) and Screw Loss (Bottom) metrics for real-world objects when the network was trained solely using simulated data [11] (lower values are better) **(b)** Predicted concentrations over articulation model parameters. DUST-net estimation performance on simulated data [11] (hatched green) included for comparison. DUST-net reported lower confidence in its predictions for real-world objects than simulated data (b), analogous to its degraded estimation accuracy(a).

## 5.3 Sim to Real Transfer

Lastly, we evaluated how effectively DUST-net transfers from simulation to a real-world setting. DUST-net was trained solely on the simulated articulated object dataset [11]. Afterward, we used it to infer the articulation model parameters for three real-world objects. Results (Fig. 5(a)) report that DUST-net outperforms the current state-of-the-art method, ScrewNet, in estimating the model parameters for real-world objects. However, the estimated parameters using DUST-net are not yet accurate enough to be used directly for manipulating these objects. This sub-par performance stems from the significant differences between the training (clean and information-rich simulation data) and test datasets, which consists of noisy depth images acquired with a Kinect sensor and contain high salt-and-pepper noise, spurious features, and incomplete objects. Better performances could be achieved by either fine-tuning the network on real-world data or retraining it using a larger real-world dataset. A noteworthy insight from the results is that DUST-net also reports low confidence over the predicted parameters for real-world objects, compared to when tested on the simulated data (Fig. 5(b)). This clearly delineates why it is beneficial to estimate a distribution over the articulation model parameters instead of point estimates. Given only point estimates of articulation model parameters, a robot has no way to determine if the estimates are reliable for manipulating the object safely or not. In contrast, DUST-net's reported confidence over the predictions could allow the robot to develop safe motion policies for articulated objects [1, 18] or use active learning based methods [17] to reduce uncertainty over the articulation parameters.

## 6 Conclusion

We introduced DUST-net, which utilizes a novel distribution over screw transforms on a Stiefel manifold to perform category-independent articulation model estimation with uncertainty estimates. We evaluated our approach on two benchmarking datasets and three real-world objects and compared its performance with two current state-of-the-art methods [11, 14]. Results show that DUST-net can estimate articulation models, their parameters, and model uncertainty estimates for novel objects across articulation model categories successfully with better accuracy than the state-of-the-art methods. At present, DUST-net can only predict parameters for 1-DOF articulation models directly. For multi-DoF objects, an additional image segmentation step is required to mask out all non-relevant object parts. This procedure can be repeated iteratively for all object part pairs to estimate relative models between object parts that can be combined later to construct a complete kinematic model for the object [10]. An interesting extension of DUST-net could estimate parameters for multi-DoF objects directly by learning a segmentation network along with it. Another exciting direction of future work is to use DUST-net in an active learning setting where, if the robot is not confident enough about the estimates of the articulation model parameters, it can actively take information-gathering actions to reduce uncertainty.

**Acknowledgments**

This work has taken place in the Personal Autonomous Robotics Lab (PeARL) at The University of Texas at Austin. PeARL research is supported in part by the NSF (IIS-1724157, IIS-1638107, IIS-1749204, IIS-1925082), ONR (N00014-18-2243), AFOSR (FA9550-20-1-0077), and ARO (78372-CS).

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
