# OpenReview forum: "Distributional Depth-Based Estimation of Object Articulation Models"
_robot-learning.org/CoRL/2021/Conference — CoRL2021 Poster_

### Official Review · Reviewer_ahNp · 2021-07-20

**Originality:** Very Good
**Technical Quality:** Very Good
**Clarity Of Presentation:** Fair
**Impact:** 3

**Recommendation:**

Weak Accept: I recommend accepting the paper, but will not argue for my recommendation if the majority of other reviewers have a different opinion.

**Summary:**

This work considers the task of predicting the screw parameters of a singly-articulated object directly from a sequence of images, as well as predicting the current rotation and displacement of the joint in each image. This specific task was introduced by ScrewNet (Jain et. al., 2020). In contrast with previous work, this work proposes to learn a probability distribution over the screw parameters, rather than a direct regression of a point estimate. Previous works in other settings have proposed distributions over screw parameters (for instance, the Bingham distribution), but such distributions suffer from the double-cover problem (or more generally, “non-unique estimation targets”). To compensate for this, the work proposes a novel parameterization of Plucker coordinates that evades the double-cover problem. Their parameterization can be factorized into a point on a Stiefel manifold and a set of positive integers, over which they can further parameterize well-studied distributions (von Mises-Fisher distribution and truncated Gaussian distributions, respectively). With this parameterization in hand, the work proposes a standard neural network architecture and training procedure to learn a distribution based on sequences of data drawn from three settings: an objects dataset from Abbatematteo et al, PartNet-Mobility, and a real-world dataset. All settings contain household objects with relatively similar door/drawer structures. When compared against SOTA methods and reasonable probabilistic baselines in noise-free settings, the work demonstrates strong performance on point estimation metrics for various components of the predicted parameter space. Furthermore, the paper convincingly demonstrates the ability to capture uncertainty over predicted parameters, particularly in settings where noise is deliberately introduced to labels at training time.

**Issues:**

- Either give a rationale for the independence assumption made by the distribution design, update the distribution design to take potential covariance into account, or state this clearly as a limitation of the proposed method.
- Improve the clarity and impact of the results section (as detailed above).


**Reviewer Expertise:**

Good: General knowledge of the area

**Strengths And Weaknesses:**

### Strengths

**Principled approach:**
The paper does a good job of considering the parameterization of distributions over Plucker coordinates, highlighting the substantial downsides of previous approaches, and proposing a principled factorization of the screw coordinates to mitigate these downsides. They choose a particularly well-understood distribution (von Mises-Fisher), which allows them to make intuitive choices for architecture and training without sacrificing theoretical soundness. Many papers fail to carefully consider the difficulties of parameterizing these structures in such a way that a neural network can learn them; the proposed method tackles the problem in a principled fashion.

**Novelty:**
A major positive aspect of this paper is its novelty and generality. While other works have variously proposed probabilistic representations of articulations, and developed category-independent articulation prediction models, this paper is the first I am aware of that unifies deep probabilistic articulation prediction by proposing 1) a parameterization that is common to all model categories, 2) a principled distribution over that parameterization, and 3) a neural network model and training algorithm that learns to predict parameters of this distribution directly from depth images.

### Major Weaknesses:

**Assumptions not clearly stated:**
The paper implicitly assumes that the probability distributions over regions of the screw space (l and m-hat; |m|) are independent; the paper also implicitly assumes that the individual displacement estimates for each input image are independent; these assumptions should be explicitly stated.

**Independence assumptions are not valid:**
This work assumes (implicitly) that an independent factorization of the probability distributions is an appropriate representation of the label space. Particularly troubling is the assumed independence between the screw parameters and the observed displacement/rotation values; presumably, these are highly dependent. Consider a case where there is uncertainty between whether the articulation is revolute or prismatic; the displacement/rotation parameters will be quite different in these cases, and so there is naturally significant covariance of these properties. Similarly, treatment of the sequence of displacement/rotation parameters as IID is likely unrealistic, given the sequential structure of the dataset (presumably watching videos of the object actuate), and joint limits present for an object. The work should either 1) consider restructuring the probability distributions to account for possible covariance, 2) demonstrate that the neural network can adequately account for these potential covariances implicitly 3) make an explicit argument for why this independence assumption is sufficient for the task, or 4) State explicitly that this is a limitation of the proposed method.

**Unclear presentation of results:**
The results are not clearly presented, and the significance of each metric is not clearly explained. It is difficult to interpret the duplicated columns, and it is difficult to interpret when higher/lower metrics are better (I recommend putting up / down arrows next to each metric to indicate this). Since the proposed method does not outperform all other methods across the board, it’s difficult to interpret which of the metrics are more important than the others. Furthermore, each table should have a detailed description of each column, and for readability, table captions should have a short interpretation or discussion of the results in the table.


### Minor weaknesses:

Missing experiment:
It would be interesting to include an ablation of DUST-Net in a local frame to see if further improvements can be obtained.

Missing precise evaluation of uncertainty:
When demonstrating that the network captures uncertainty (Sec 5.2), the paper only shows a monotonic correlation between input noise and certainty; no quantitative comparison is given (e.g. comparing the noise distribution with the uncertainty distribution, which may be possible since both distributions are closed form)

In the parameterization, why are d and theta considered independently at each timestep, instead of predicting the pitch h of the articulation for the whole sequence and a single h for the sequence?

Using a “ReLU-6” module in the network seems to be imprecise, since the purported limit for the parameters is 2pi, not 6.

It would be helpful to perform an ablation on the training procedure to show the benefit of their three-stage approach.

How does the work arrive at the conclusion that the vm-SoftOrtho suffers because of the soft constraint? Could the work include some measure of how much orthogonality is violated, for typical predictions?

In the uncertainty estimation section, it would be helpful to explain in more detail what the uncertainty parameters mean, particularly in the summary tables. A clearer way to present this information is by giving an estimate of the 95th-percentile confidence intervals and aggregating in some way, to ground the metrics intuitively.

For the ScrewNet performance in Table 4, is this in “local” or “camera frame” coordinates?

This paper would be stronger if it included an example of how to use the knowledge of this uncertainty in a downstream task. For instance, the paper doesn’t give a procedure for how to induce a posterior over this distribution should new observations be obtained.

The presentation of the Preliminaries (Sec 3) is quite abstract; the presentation would benefit by first grounding both the distribution and the manifolds in the practical instances (i.e. V_2,3), which is easier to grasp intuitively than V_n,k. It took a close reading to understand how the abstract form related to the problem domain.

If possible, a visualization of the uncertainty over multiple examples would lend a lot of clarity to the performance of the method.

Typo:
121: “prismatic” is repeated twice


**Summary Of Recommendation:**

As far as I can tell, this paper is the first to present a method for learning a probability distribution over screw parameters.  The proposed approach demonstrates superior performance on several metrics across tasks that are directly relevant to robotics. As such, I believe it meets the minimum standard for acceptance. However, I am hesitant to give a strong accept, because:
- No discussion/reasoning for the independence assumptions of their proposed distribution is provided. This undermines the ability for the method to claim to be entirely principled.
- The presentation of the results is not clear (both the text and the design of the tables), and no intuition is given for how to interpret the numerical performance on each of the provided metrics. Why are the metrics that the proposed method performs well on more important than the metrics in which it doesn’t?
- The actual utility of modeling uncertainty is not demonstrated. Not enough motivation is given on how this information could be used in a downstream task.

# Post-rebuttal Update
I thank the authors for making the suggested changes. I believe that the clarity of the paper has improved. Most of my questions have been sufficiently resolved.

However, I still think the limitations of the distribution approximation (discussed in the initial review and the rebuttal) should be more clearly stated.

> To maintain the numerical tractability of the solution, we approximate the probability density function of the conditional distribution as a Dirac delta function centered at the expected value of the distribution over the screw axis parameters. We have included further details and discussion on the topic in the revised supplementary material for the work.

This approximation could have a significant effect on the result; I believe that it should be mentioned (at least briefly) in the main text (and then refer the reader to the supplement for additional details). For example, if the variance of the screw parameters is relatively high, then there will be much probability mass far from the expected value, and this approximation will likely be poor. This is fine, but the approximation should be explained clearly to the reader in the main text.

Overall, looking at the author response and the other reviews, my score remains unchanged.

---

> ### Author Response · Authors · 2021-08-28
> **Reply to Reviewer ahNp**
>
> We thank the reviewer ahNp for their detailed feedback. We would like to address the concerns raised by them as follows:
> - **Stating modeling assumptions**: We thank the reviewer for their help in improving the quality of the manuscript. We have revised our manuscript and the supplementary material to clearly state and justify the assumptions we make while modeling the joint distribution over the articulation model parameters.
>
> - **Validity of assumptions**: We thank the reviewer for their insightful comment. We first address the assumptions we make to define the distribution over the screw axis and the joint configuration parameters. We would like to clarify that we do not assume independence between the screw axis parameters and the observed joint configurations. Instead, we define the joint distribution over the articulation model parameters, <S, \theta_{1:n-1}, \d_{1:n-1}>, as a product of a distribution over the screw axis parameters and a conditional distribution over the joint configuration parameters. We first approximate the distribution over the screw axis parameters, S, as a product of two marginal distributions, one over the orientation vector tuple and another over the moment vector magnitude, using a matrix von Mises-Fisher distribution over Stiefel manifold and a truncated normal distribution over the set of positive real numbers. This approximation is motivated by the fact that calculating statistics over manifolds can be computationally intractable in a general setting [1],[2],[3]. It enables us to define the probability density function over the screw axis parameters using standard distributions over manifolds whose properties are well studied in the literature, such as the matrix von Mises-Fisher distributions over Stiefel manifolds [1],[2].
>
>      Calculating the conditional distribution over joint configurations exactly would require us to evaluate hypergeometric functions over the complete manifold in which the screw transforms lie. Hypergeometric functions in the matrix argument result in an infinite series in terms of zonal polynomials, which becomes combinatorially expensive to calculate with the increasing number of terms [3]. To maintain the numerical tractability of the solution, we approximate the probability density function of the conditional distribution as a Dirac delta function centered at the expected value of the distribution over the screw axis parameters. We have included further details and discussion on the topic in the revised supplementary material for the work.
>
>     Regarding the assumptions over the sequence of joint configuration parameters (displacement/rotation parameters), we would like to begin by noting that the unified parameterization of the articulation model parameters corresponds to a sequence of rigid body transforms (or screw transforms). Each of these rigid body transforms can be treated as an independent frame transformation between the object parts. Leveraging this fact, we approximate the conditional distribution over the joint configurations as a product of marginals over screw transforms at each time step, leading to the particular parameterization presented in the manuscript.
>
>    We would like to note again that we have revised our manuscript and the supplementary material to clearly state and justify the assumptions we make while modeling the joint distribution over the articulation model parameters.
>
> - **Presentation of results**: We thank the reviewer for their help in improving the quality of the manuscript. We have updated the manuscript to include plots comparing the performance of DUST-net with other methods for all the experiments. We have also updated the discussions over experimental results to further clarify their implications.
>
> - **Evaluation of uncertainty**:  We wish to clarify that we refrained from including further quantitative comparisons between the input noise and the predicted uncertainty from DUST-Net in the text primarily because we make strong simplifying assumptions while modeling the input noise to the system. The sources of uncertainty in real-world data are far more complex and sophisticated than the input noise model we considered in this work, especially for the set of experiments discussed in section 5.2. Due to these significant differences, we believe that the numerical analysis of these results is not appropriately informative. Instead, we chose to present the qualitative results that demonstrate that the predicted uncertainty from DUST-Net follows a similar trend as in the input noise. We would also like to note that for a downstream robot manipulation task of interacting with articulated objects safely, the availability of indicative but ordered uncertainty estimates may be sufficient for choosing the appropriate actions [5].

---

> > ### Author Response · Authors · 2021-08-28
> > **Reply to Reviewer ahNp (Contd.)**
> >
> > - **Joint configuration v/s pitch**:  We thank the reviewer for suggesting an alternative parameterization for representing object articulation models. While the proposed alternative parametrization helps in reducing the total number of parameters required to represent the articulation model, it may not be a suitable representation for training the network. This primarily stems from the fact that the proposed parameterization results in extreme values, 0 and infinity, for the pitch of two articulation model categories, prismatic and revolute, respectively. These extreme values as targets can destabilize the training of a network that is learning to estimate the articulation model properties of objects across articulation model categories, which is the case for DUST-Net.
> >
> > - **ReLU-6**: We wish to clarify that we do scale back the output of the ReLU-6 layer to the appropriate range of the parameters, be it pi or 2pi as discussed in the main text. We used the ReLU-6 layer instead of a Hardtanh layer (range [-1, 1]) primarily to increase the precision in the network predictions [4].
> >
> > - **vm-SoftOrtho soft constraint violation**: We conclude that vm-SoftOrtho suffers because of the soft constraint as results for vm-SoftOrtho over the screw loss metric were obtained after imposing the hard orthogonality constraint on its predictions, and we observed that its performance degraded significantly. We thank the reviewer for suggesting that including a measure on the degree of violation of the orthogonality constraint by vm-SoftOrtho’s predictions will enhance the quality of the manuscript. We will include these results in the supplementary material of the work.
> >
> > - **Clarifications on Table 4**: ScrewNet’s performance in table 4 is reported in the “camera-frame” coordinates.
> >
> > - **Examples of downstream tasks**: As we noted in the text, the availability of uncertainty estimates over articulation model parameters will open avenues for a wide variety of downstream tasks involving articulated objects. Some specific examples are as follows:
> >   - It enables the learning of kinematic models for articulated objects with model uncertainty estimates directly from visual data. With an estimate of model uncertainty, a POMDP planner can be used to develop safe motion plans to interact with articulated objects.
> >   - Uncertainty estimates over articulation model parameters can also facilitate learning robot behavior policies for interactions with articulated objects directly from data. They can guide the exploration while learning robot behavior policies that provide safety guarantees [5].
> >   - Uncertainty estimates predicted by DUST-Net indicate the network’s confidence over the estimated articulation model parameters for objects. Hence, they can be used in conjunction with an active learning-based approach, such as [6], to take information gathering actions for acquiring information-rich data so as to increase the network’s confidence over the estimated parameters.
> >
> >
> >
> > **References**:
> > - [1] Y. Chikuse.Statistics on special manifolds, volume 174. Springer Science & Business Media, 2003. doi:https://doi.org/10.1007/978-0-387-21540-2
> > - [2] K. V. Mardia and P. E. Jupp.Directional statistics, volume 494.  John Wiley & Sons, 1999. doi:10.1002/9780470316979.
> > - [3] L. Jiu and C. Koutschan.  Calculation and properties of zonal polynomials. Mathematics in Computer Science, pages 1–18, 2020
> > - [4] A. G. Howard, M. Zhu, B. Chen, D. Kalenichenko, W. Wang, T. Weyand, M. Andreetto, and H. Adam. Mobilenets: Efficient convolutional neural networks for mobile vision applications. arXiv preprint arXiv:1704.04861, 2017
> > - [5] A. Taylor, A. Singletary, Y. Yue, and A. Ames.  Learning for safety-critical control with control barrier functions.  InProceedings of the 2nd Conference on Learning for Dynamics and Control, 2020.
> > - [6] Y. Cui and S. Niekum.  Active reward learning from critiques.   In 2018 IEEE International Conference on Robotics and Automation (ICRA), pages 6907–6914. IEEE, 2018.

---

> > > ### Comment · Reviewer_ahNp · 2021-09-03
> > > **Post-rebuttal update**
> > >
> > > I thank the authors for making the suggested changes. I believe that the clarity of the paper has improved. Most of my questions have been sufficiently resolved.
> > >
> > > However, I still think the limitations of the distribution approximation (discussed in the initial review and the rebuttal) should be more clearly stated.
> > >
> > > > To maintain the numerical tractability of the solution, we approximate the probability density function of the conditional distribution as a Dirac delta function centered at the expected value of the distribution over the screw axis parameters. We have included further details and discussion on the topic in the revised supplementary material for the work.
> > >
> > > This approximation could have a significant effect on the result; I believe that it should be mentioned (at least briefly) in the main text (and then refer the reader to the supplement for additional details). For example, if the variance of the screw parameters is relatively high, then there will be much probability mass far from the expected value, and this approximation will likely be poor. This is fine, but the approximation should be explained clearly to the reader in the main text.
> > >
> > > Overall, looking at the author response and the other reviews, my score remains unchanged.

---

### Official Review · Reviewer_41iG · 2021-07-23

**Originality:** Very Good
**Technical Quality:** Good
**Clarity Of Presentation:** Very Good
**Impact:** 4

**Recommendation:**

Weak Accept: I recommend accepting the paper, but will not argue for my recommendation if the majority of other reviewers have a different opinion.

**Summary:**

The paper proposes DUST-net, a deep-learning approach that estimates distributions over articulation models and articulated pose.
The model is object-category agnostic, using screw theory to represent revolute & prismatic joints with the same model.
A novel distribution over screw axes is developed using von Mises-Fisher distributions on Stiefel manifolds which cleanly resolves issues with existing parameterizations and enables data-efficient learning.
The experiments are thorough with thoughtful baselines and ablations.


**Issues:**

Minor issues.
Line 92 "Guassian"
Line 195 "whit"
Line 318 "form"
Table 4 caption lacking period

**Reviewer Expertise:**

Very good: Comprehensive knowledge of the area

**Strengths And Weaknesses:**

Strengths
- The paper presents important conceptual improvement over ScrewNet in uncertainty representation, camera-frame estimation.
- The theory is quite rich and the technical approach is sound.
- The paper is generally well-written.

Weaknesses
- Motion of an object part isn't generally available in the wild - I'd recommend an experiment with static objects for a sense of how the approach would work without a human demonstrator for every object. This is a more appropriate comparison to the Abbatematteo paper.
- I'd like to see a comparison to a generic feature-tracking approach to understand how significant the textured-object limitation is in practice.
- Real results are disappointing. But, they certainly give credence to the importance of uncertainty estimation. I suspect these could improve with some care taken toward the noise model used in training, as depth images with pixel dropout have been shown to generally transfer fairly well.

- Object category generally provides a useful prior over articulation type and parameters, and is generally available in the wild. This could be readily incorporated into the model.

**Summary Of Recommendation:**

I am recommending weak acceptance. The absence of successful real results is disappointing but I think the theory is sufficiently interesting and the simulation experiments are sufficiently thorough.

---

> ### Author Response · Authors · 2021-08-28
> **Reply to Reviewer 41iG**
>
> We thank the reviewer 41iG for their time and feedback. We would like to address the concerns raised by them as follows:
> - **Static objects as input**: We thank the reviewer for suggesting an interesting experimental study on analyzing DUST-Net’s performance on static objects. While we agree that in the wild the motion of an object part may not always be readily available, we wish to clarify that DUST-Net does not necessarily require a human demonstrator for every object to estimate their articulation models. One promising future research direction that we would like to explore is to use active learning and interactive perception-based approaches to guide robot’s interactions with objects for learning their articulation models. We believe that while static visual features may be informative to learn priors over object articulation models, observations of the motion of articulated object parts are required to correctly estimate an object’s articulation model with high confidence in a general setting. It is especially true for some of the articulated objects commonly found in human environments, for example, an office desk with drawers. Due to aesthetic reasons, the door handles for all the drawers may look similar, but based on their specific design, some of the drawers may have a hinge joint instead of a sliding joint. To correctly estimate the articulation model for such a desk, a robot or a human must first interact with the object to induce motion in different object parts and then, based on the observations, estimate the articulation model types present in it.
>
> - **Comparison to a feature-tracking approach**: We thank the reviewer for suggesting an interesting baseline method to compare DUST-Net’s performance. However, conducting experiments with a texture-based feature-tracking approach will require adding textures to the objects available in the benchmarking datasets, which may not be possible to include in the current version of the manuscript due to limited time.
>
> - **Results on real-world objects**: We thank the reviewer for their suggestions. We will incorporate them to improve DUST-Net’s estimation performance on real-world objects. We are also collecting a larger real-world dataset that may be used to finetune the network for better performance on real-world objects.
>
> - **Object category prior**: We thank the reviewer for their suggestion on improving DUST-Net’s performance. However, we would like to note that one of our primary goals through the work is to eliminate the requirement of knowing an object’s articulation model category a priori for estimating its articulation model. Incorporating the proposed suggestion in DUST-Net will not serve the goal of category-independent object articulation model estimation.

---

### Official Review · Reviewer_tV4q · 2021-07-24

**Originality:** Very Good
**Technical Quality:** Good
**Clarity Of Presentation:** Fair
**Impact:** 4

**Recommendation:**

Weak Accept: I recommend accepting the paper, but will not argue for my recommendation if the majority of other reviewers have a different opinion.

**Summary:**

The paper proposes a method to learn distributions over articulation models from a sequence of depth images without any known articulation model a priori. To learn the distribution, the paper first presents a novel representation for distribution over rigid body transformations and articulation model parameters using Screw theory, von Mises-Fisher distributions and Stiefel manifolds. Then the paper introduces DUST-net, a network that transforms a sequence of depth images into a distribution over the transformations between them. With experiments on two benchmarking datasets (a simulated articulated object dataset and PartNet-Mobility dataset) and three real-world objects, it is shown that DUST-net successfully learns the distribution over articulation models and their parameters.

**Issues:**

The issues are listed above.

**Reviewer Expertise:**

Good: General knowledge of the area

**Strengths And Weaknesses:**

Strengths:
* The paper addresses an important problem of perceiving articulated objects from a sequence of observations without any prior knowledge of the object model or its articulation constraints.
* The paper attempts to present a suitable representation that can capture distribution over transformations and articulation parameters. If this representation has the benefits as claimed in the paper, then it could be used for part based object estimation frameworks.
* The presented representation lets the proposed method perform category-independent articulation model estimation along with model uncertainties.

Questions and areas for improvement:
* In Section 3, especially the background section needs to be closely knit with the representation of an articulated model. Currently, it is hard for the reader to relate these concepts with the problem this paper is trying to address.
* The relationship between Screw Transformations, Stiefel manifold and Matrix von Mises-Fisher distribution is unclear at Section 3. The fundamental background of these concepts can be in the supplementary material, while their relationship and relevance to the representation of articulated models can be described in the main manuscript.
* It is unclear what Figure 2 is trying to illustrate.
* Visuals or illustrations to support the description in Section 4 would be beneficial.
* In Tables 1, 2, and 3, it is unclear what “Uncertainty” values indicate. It is described in the text that the uncertainty is high on real-world experiments, whereas the numbers are less in Table 3 compared to 1 and 2.
* In Section 5.2, the text - “We evaluate this by adding artificial noise to the training labels form the two simulated datasets while training DUST-net”. It is not clear how training with noise leads to this evaluation. Is this a different training as opposed to the original for estimating the articulation models parameters? The reader would imagine that there is one training phase overall, that includes the addition of noise to the labels.
* It is described in the Section 5.3 that the estimation is accurate in the real-world setting. How will the proposed approach handle partial observations caused due to self-occlusions and robot occlusions during the manipulation action, in addition to the depth noise from the sensor data? Is it possible to characterize the uncertainty due to sensing and the scenario differently?

Weakness:
* Main concern with the current version of the manuscript is that it is hard to follow. The paper is overloaded with concepts that require a background. Supplementary material could be used to develop basic understanding of the preliminaries presented in section 3, while the section 3 in itself can relate these concepts to the representation of an articulated model.
* No visuals to support sections 3 and 4.
* No qualitative examples of the articulated objects along with their model estimates and uncertainties are provided to understand the results.


**Summary Of Recommendation:**

The paper presents a novel representation for distribution over rigid body transformations and articulation model parameters. The paper attempts to describe the theoretical background in developing this background. However, the major concern with the current version of the paper is that it is dense and packed with concepts that require background. This makes it hard to judge the claims of the paper. The quantitative results presented are promising. However, the lack of qualitative results is a weakness.

---

> ### Author Response · Authors · 2021-08-28
> **Response to Reviewer tV4q**
>
> We thank the reviewer tV4q for their time and feedback. We would like to address the concerns raised by them as follows:
> - **Background section and visualization of experimental results**: We thank the reviewer for their help in improving the quality of the manuscript through insightful feedback. We have revised our manuscript to reflect the suggestions and feedback provided.
>
> - **Uncertainty estimation experiments clarifications**: We would like to clarify that for the experiments discussed in section 5.2, we considered a different training procedure than the one we used to train DUST-Net for other experiments. In that particular set of experiments, we created a new training dataset for each choice of the artificial input noise and trained a separate network. The reported results present the correlation between the injected noise and the precision parameters of the predicted distributions by the networks trained with the injected noise on the test dataset. We will revise the manuscript to clarify it further in our discussion.
>
> - **Noise due to occlusions and depth noise**: We would like to clarify that the training and the test datasets we considered included instances with varying degrees of object self-occlusions. We also included object instances when the object may only be partially visible from the camera’s viewpoint to simulate robot occlusions during robot-object interactions.
>
> To simulate the noise present in the real-world depth data, we considered the pixel-dropping noise model, which probabilistically drops the image pixels to simulate the problem of pixel dropouts commonly found in the depth data acquired using an RGB-D sensor such as a Kinect sensor.
>
> Separately characterizing the uncertainty due to sensor noise and due to occlusions in the visual data is a challenging task. While we appreciate its importance in understanding the sources of uncertainty in the input data, we believe that it is beyond the scope of the work.

---

> > ### Comment · Reviewer_tV4q · 2021-09-02
> > **Reviewer response**
> >
> > Thanks to the authors for answering some of my questions and updating the submission. I will maintain my original score.

---

### Official Review · Reviewer_Libh · 2021-07-27

**Originality:** Good
**Technical Quality:** Very Good
**Clarity Of Presentation:** Good
**Impact:** 3

**Recommendation:**

Weak Reject: I recommend rejecting the paper, but will not argue for my recommendation if the majority of other reviewers have a different opinion.

**Summary:**

Given a sequence of depth images depicting an object undergoing an articulation motion, this paper tackles the task of predicting: a) a parametrization of the articulation joint, and b) the per-timestep configuration. It does so by building upon a prior supervised approach "ScrewNet" that presents a unified representation (screw axis) which allows capturing various joint types.

This approach leverages a similar representation, supervision, architecture, with the key contribution being to predict a distribution instead of a point estimate for the screw axes and corresponding per-timestep parameters. The distribution over the screw axis is parametrized as a von mises fisher distribution, with further factorization in predicting the matrix. The other variables are all represented via truncated gaussian distributions.

The reported experiments show that this improves the performance over the baseline while also allowing capturing uncertainty in the predictions.

**Issues:**

I have some minor concerns/clarifications regarding the precise formulations and task definitions:
a) The per-timestep parameters \theta and \d are modeled via truncated gaussians, but I am not sure why these can't have negative values e.g. if a joint moves back and forth around an initial configuration?

b) While the training objective is to maximize the likelihood of the ground-truth, I am slightly confused by how a unique screw axis is unambiguously defined as the ground-truth e.g. for a prismatic joint, wouldn't any axis parallel to the part motion suffice?

c) I appreciate that both theta and d need to be predicted for a unified representation, but isn't only one of them non-zero depending on the joint type (e.g. in Figure 1)? I understand that a helical motion for example max exhibit both, but given the paper only considers revolute and prismatic joints, only one of these could be non-zero in predictions - and given the independent marginals, this is a correlation that the proposed distributions cannot model.

**Reviewer Expertise:**

Fair: Some knowledge of the area

**Strengths And Weaknesses:**

**Strengths**

a) The paper is well-motivated. The recent prior work that tackles this task via a learning-based method only predicts a point-estimate, and extending it to allow capturing the uncertainty is a promising directions.


b) The empirical results are (mostly) convincing and demonstrate clear improvements from the proposed approach over the relevant baselines.

c) Additional ablations reported e.g for directly predicting 'F' for the VMF distribution vs the proposed factorization via 4 angles and a diagonal matrix is also interesting and validates the specific choices made.


**Weaknesses**

- While capturing uncertainty is desirable, I am not sure the proposed approach is a significant contribution on its own - in particular given that the way the distribution is represented is rather straightforward (except for the screw axis) and the work is a direct extension of ScrewNet.

- As a related point, while the approach does enable representing some distributions, I am concerned about the limited expressivity as:
a) that the proposed approach still only captures a unimodal distribution whereas often they might be multimodal e.g. if one is unsure of joint type (revolute vs prismatic) but sure of geometry, the screw axes may have different modes.
b) The distribution for the per-timestep parameters is also modeled as a product of independent marginal terms and thus does not represent possible correlation among them.
Overall, for a work explicitly aimed at modeling the uncertainty in the representations, I feel this work only takes small initial step.

- A minor concern regarding the evaluations Tables 1,2 is that they claim a higher precision i.e lambda_i, and \beta_j is desirable. However, it is not clear why they should be arbitrarily high even for non-noisy training data (perhaps it maybe only possible to recover them upto a certain degree given perceptual inputs). I think these estimates could be removed from these Tables as it is unclear if higher is better.


----
Update:
I’d like to thank the authors for their response, and in particular for the explanation of contributions over ScrewNet. However, I am still not convinced regarding the generality of the proposed uncertainty representation. Moreover, some of the responses to the implementation “issues” further increased this feeling (e.g. an adhoc procedure for deciding a unique “ground-truth” screw axis when multiple solutions are acceptable seems suboptimal).

But more importantly, I am not convinced the independent inference of model parameters and screw transform is a benefit. The response claims that “in the cases when there are ambiguities in the object's articulation type, DUST-Net will predict distributions over both observed joint configurations to be centered at non-zero values”. However, this only works because the two types of transforms considered here only affect disjoint parameters (either \theta, or \d).
Considering an extension of this approach, say including helical joints, if the model is confused between two possible axes (say s_1 and s_2), and each of the correspondingly has different transforms the model would want to infer (say theta_1, and theta_2). The proposed approach cannot handle this for two reasons: i) the prediction over theta is multimodal, and ii) more critically, this distribution is conditional on the screw transform (e.g. theta_1 for s_1, and theta_2 for s_2).


**Summary Of Recommendation:**

Overall, while the work is well-motivated in aiming to predict a distribution and also yields empirical gains, I am not excited regarding the technical contributions and also skeptical about the generality of the distributions (as they cannot capture multimodal outputs).

---

> ### Author Response · Authors · 2021-08-28
> **Response to Reviewer Libh**
>
> We thank the reviewer Libh for their time and feedback. We would like to address the concerns raised by them as follows:
> - **Contribution Impact**: We would like to clarify that though our work uses the unified representation proposed in the prior work “ScrewNet'' to represent the articulation models, our work addresses several non-trivial challenges that ScrewNet struggles with: 1) DUST-net predicts distributions over articulation model parameters, whereas ScrewNet can only predict their point estimates, 2) DUST-net learns such distributions in a probabilistic learning setting, unlike ScrewNet, which helps DUST-Net to be more robust to outliers and noise than ScrewNet, 3) DUST-Net is trained using a well-understood convex training loss function, the log-likelihood function, whereas, ScrewNet is trained using a complex loss function with multiple competing objectives that can be harder to train, 4) DUST-Net's network architecture is improved from that of ScrewNet. ScrewNet uses an additional LSTM layer to impose temporal dependence between the input sequence of images. DUST-Net does not require these layers, which helps it to be more data-efficient than ScrewNet while reporting superior performance.
>
>     Moreover, we would like to note that choosing an appropriate distribution over the articulation models is a challenging task. One of the ablation methods we compare DUST-Net with, the vm-SoftOrtho method can be seen as an extension of ScrewNet to the distributional setting. It is evident from the experimental results that it leads to subpar performance compared to DUST-Net. This reinforces that choosing an appropriate distribution over the articulation models is not a straightforward task but requires careful consideration of the inherent properties of the screw transforms.
>
> - **Multimodal distribution**: We wish to clarify that the specific example considered by the reviewer will not pose any major challenges to our approach as DUST-Net learns distributions over screw transforms rather than over the object articulation models directly. As a screw transform can represent all possible combinations of rotations and displacements that may occur about the screw axis, DUST-Net does not need to know the object articulation model category at the inference time. Under this choice of representation, if the network is confident about the object's articulation type, it will predict distributions over one of the joint configurations to be centered at values significantly greater than zero and those for the other configuration parameter to be concentrated primarily at zero. For example, if the network is confident that the object contains a hinge joint, it will predict distributions over the observed rotations, \theta, to be centered at values substantially greater than zero and those for the observed displacements, \d, to be concentrated around zero. Analogously, in the cases when there are ambiguities in the object's articulation type, DUST-Net will predict distributions over both observed joint configurations to be centered at non-zero values.
>
>     Additionally, we would like to note that the current framework does not impose any restrictions on the choice of distributions to use. In our experiments, we used unimodal distributions and still found DUST-Net to be more expressive and better performing than other state-of-the-art methods. It indicates that estimating articulation models using a distributional approach has far higher representational power than the methods that predict point estimates, even when the distributional approach makes simplifying assumptions. In future work, DUST-Net could be extended to use a multimodal distribution over the articulation model parameters, defined using a mixture of matrix von Mises-Fisher distributions over the Stiefel manifold and mixtures of truncated Gaussians over other parameters.
>
> - **Independence Assumption for per-time step parameters**: We agree that as the distribution over the per-timestep joint configurations, \theta, and \d, is defined as a product of independent marginal distributions, the proposed distribution can not represent the possible correlations between them. We made the independence assumption between the joint configurations to simplify the network’s training procedure. However, we would like to note that this assumption did not affect DUST-Net’s performance markedly, as DUST-Net still outperformed the other state-of-the-art methods while being both more data and computationally efficient. In future work, we plan to address this assumption by using a multivariate Gaussian distribution to represent the distribution over the per-timestep joint configurations; however, this work is beyond the scope of this paper.
> We have also revised the supplementary material to include a detailed discussion on the assumptions that we make in the work. We would request the reviewer to please refer to it for further justifications.

---

> > ### Author Response · Authors · 2021-08-28
> > **Response to Reviewer Libh (Contd.)**
> >
> > - **Non-negative values for \theta and \d**:  We would like to clarify that the choice of non-negative values of the joint configuration variables \theta and \d is a design choice that we made to remove ambiguous estimation targets for the network. In the specific example considered by the reviewer, i.e., “if a joint moves back and forth around an initial configuration”, without this restriction, a forward motion about the joint can be treated as both a forward motion about the axis oriented along the “+l” vector and a backward motion about the axis oriented along the “-l” vector. It results in a non-unique estimation target for the estimator and can hinder the estimator's performance. DUST-Net overcomes this problem by estimating the forward and the backward motion about the joint as motion about two separate screw axes, one oriented along the "+l" vector and the other along "-l" vector, with both having non-negative joint configuration values. More detailed justifications behind this particular choice along with other design choices were presented in section 4 of the manuscript and the supplementary text. We will further expound our discussions in the manuscript to explain the specific design choices clearly in the camera-ready version of the paper.
> >
> >
> > - **Unique prismatic screw axis**: We address the problem of defining unique screw axes for prismatic joints by making specific design choices in the training dataset. We consider the screw axis for a prismatic joint to pass through the center of the bounding box of the moving part of the object while generating the training dataset.
> >
> >
> > - **Correlation between non-zero joint configuration and object joint type**: We would like to clarify that our choice of using a unified representation for the articulation model parameters was made specifically to avoid the correlation between the object joint type and the model parameters. This particular correlation requires knowing the type of the joint present in the object a priori for estimating which of the two joint configuration parameters is non-zero. In a general setting, a robot will not have access to this information about the objects that it will interact with beforehand. Hence, having a model parameterization dependent upon the availability of this information is a strong assumption, which this work aims to alleviate.

---

### Meta-Review · Area_Chair_5hxH · 2021-08-16

**Recommendation:** Accept (Poster)
**Confidence:** 3

**Metareview:**

According to Reviewer ahNp, this is the first work to present a method for learning a probability distribution over screw parameters. The paper chooses a mathematically principled distribution (VMF) to represent a distribution over screw parameters without ambiguity. The paper empirically motivates a distributional approach over a point estimate of screws.

Two of the reviewers believe the paper to have new ideas that will be relevant to robotics / ML, while two reviewers believe the work to be technically correct but incremental. In particular, Reviewer ahNp raises three major issues required to lift their rating, which I am inclined to agree with:
```
- No discussion/reasoning for the independence assumptions of their proposed distribution is provided. This undermines the ability for the method to claim to be entirely principled.
- The presentation of the results is not clear (both the text and the design of the tables), and no intuition is given for how to interpret the numerical performance on each of the provided metrics. Why are the metrics that the proposed method performs well on more important than the metrics in which it doesn’t?
- The actual utility of modeling uncertainty is not demonstrated. Not enough motivation is given on how this information could be used in a downstream task.
```
I would also like the authors to address the questions in the "Issues" section brought up by Reviewer Libh.

This paper does not apply distributional depth-based articulation estimates to an actual robotic control task, so it isn't obvious to the reviewers and myself what the utility of the distributional representation is until we see it on the robot. I will note that there was a concurrent work submitted (https://openreview.net/forum?id=bEito8UUUmf) that shows that uncertainty-aware refinement of depth predictions does lead to substantial performance improvements on a detection task, so perhaps something similar might go on here. The authors should suggest in the paper how exactly the distributional representation of the screw predictions can be utilized to confer benefits over point estimates of screws. For example, a distributional representation might allow for running a test-time MCMC sampling procedure to resolve the covariate issue raised by ahNp, as one way to address the modeling of co-varying screws.

---- 20210904

The authors have addressed my concerns and the majority of reviewers now vote to accept, so I will be recommending an accept. I think broadly all the reviewers agree that connecting the distributional representations to an actual robotics task is the next step in this work.

---

> ### Author Response · Authors · 2021-08-28
> **Reply to Area Chair 5hxH**
>
> We thank the area chair for their time and for providing us with such meaningful feedback. We would like to address the concerns raised as follows:
> - **Independence assumptions**: We have revised our manuscript to clearly state and justify the assumptions we make while modeling the joint distribution over the articulation model parameters. We have also briefly described them in our response to the reviewer ahNp.
> - **Better presentation of results**: We have updated the manuscript to include plots comparing the performance of DUST-net with other methods for all the experiments. We have also updated the discussions over experimental results to further clarify their implications.
> - **Utility of modeling uncertainty**: We thank the meta-reviewer for suggesting a possible use case scenario of our method. Modeling uncertainty over articulation model parameters opens avenues for various other downstream tasks as well, such as robust motion planning to interact with articulation objects safely and using active learning based approaches to gather information-rich data for better estimation of model parameters. We provide further concrete examples in our response to the reviewer ahNp.
> - **Other concerns**: We have added our responses to the specific concerns raised by the reviewers as a reply to their comments.

---

### Decision · Program_Chairs · 2021-09-13

**Decision:**

Accept (Poster)

**Comment:**

According to Reviewer ahNp, this is the first work to present a method for learning a probability distribution over screw parameters. The paper chooses a mathematically principled distribution (VMF) to represent a distribution over screw parameters without ambiguity. The paper empirically motivates a distributional approach over a point estimate of screws.

Two of the reviewers believe the paper to have new ideas that will be relevant to robotics / ML, while two reviewers believe the work to be technically correct but incremental. In particular, Reviewer ahNp raises three major issues required to lift their rating, which I am inclined to agree with:
```
- No discussion/reasoning for the independence assumptions of their proposed distribution is provided. This undermines the ability for the method to claim to be entirely principled.
- The presentation of the results is not clear (both the text and the design of the tables), and no intuition is given for how to interpret the numerical performance on each of the provided metrics. Why are the metrics that the proposed method performs well on more important than the metrics in which it doesn’t?
- The actual utility of modeling uncertainty is not demonstrated. Not enough motivation is given on how this information could be used in a downstream task.
```
I would also like the authors to address the questions in the "Issues" section brought up by Reviewer Libh.

This paper does not apply distributional depth-based articulation estimates to an actual robotic control task, so it isn't obvious to the reviewers and myself what the utility of the distributional representation is until we see it on the robot. I will note that there was a concurrent work submitted (https://openreview.net/forum?id=bEito8UUUmf) that shows that uncertainty-aware refinement of depth predictions does lead to substantial performance improvements on a detection task, so perhaps something similar might go on here. The authors should suggest in the paper how exactly the distributional representation of the screw predictions can be utilized to confer benefits over point estimates of screws. For example, a distributional representation might allow for running a test-time MCMC sampling procedure to resolve the covariate issue raised by ahNp, as one way to address the modeling of co-varying screws.

---- 20210904

The authors have addressed my concerns and the majority of reviewers now vote to accept, so I will be recommending an accept. I think broadly all the reviewers agree that connecting the distributional representations to an actual robotics task is the next step in this work.